Nontargeted and targeted metabolomics approaches reveal the key amino acid alterations involved in multiple myeloma

Yue Lingling 1
Zeng Pengyun 1
Li Yanhong 2
Chai Ye 1
Wu Chongyang 1
Gao Bingren bingrengao@foxmail.com 3
1 Department of Hematology, Lanzhou University Second Hospital , Lanzhou , Gansu , China
2 Institute of Hematology, Lanzhou University Second Hospital , Lanzhou , Gansu , China
3 Department of Cardiac Surgery, Lanzhou University Second Hospital , Lanzhou , Gansu , China
Sharma Gaurav
Electronic publication date: 2022 Feb 9
Publication date: 2022
Volume: 10
Electronic Location ID: e12918
Received 2021 Jul 5; Accepted 2022 Jan 20
Copyright: ©2022 Yue et al.
Copyright year: 2022
Copyright holder: Yue et al.
License: This is an open access article distributed under the terms of the Creative Commons Attribution License, which permits unrestricted use, distribution, reproduction and adaptation in any medium and for any purpose provided that it is properly attributed. For attribution, the original author(s), title, publication source (PeerJ) and either DOI or URL of the article must be cited.
License URL: https://creativecommons.org/licenses/by/4.0/

Keywords: Multiple myeloma, Metabolomics profiles, Metabolic pathways, Differential amino acid metabolites, Diagnostic biomarkers

Funding: Cuiying Scientific and Technological Innovation Program of Lanzhou University Second Hospital CY2017-MS13 This work was supported by the Cuiying Scientific and Technological Innovation Program of Lanzhou University Second Hospital, CY2017-MS13. The funders had no role in study design, data collection and analysis, decision to publish, or preparation of the manuscript.

==============================
Purpose

Multiple myeloma (MM), a kind of malignant neoplasm of clonal plasma cells in the bone marrow, is a refractory disease. Understanding the metabolism disorders and identification of metabolomics pathways as well as key metabolites will provide new insights for exploring diagnosis and therapeutic targets of MM.

Methods

We conducted nontargeted metabolomics analysis of MM patients and normal controls (NC) using ultra-high-performance liquid chromatography (UHPLC) combined with quadrupole time-of-flight mass spectrometry (Q-TOF-MS) in 40 cases of cohort 1 subjects. The targeted metabolomics analysis of amino acids using multiple reaction monitoring-mass spectrometry (MRM-MS) was also performed in 30 cases of cohort 1 and 30 cases of cohort 2 participants, to comprehensively investigate the metabolomics disorders of MM.

Results

The nontargeted metabolomics analysis in cohort 1 indicated that there was a significant metabolic signature change between MM patients and NC. The differential metabolites were mainly enriched in metabolic pathways related to amino acid metabolism, such as protein digestion and absorption, and biosynthesis of amino acids. Further, the targeted metabolomics analysis of amino acids in both cohort 1 and cohort 2 revealed differential metabolic profiling between MM patients and NC. We identified 12 and 14 amino acid metabolites with altered abundance in MM patients compared to NC subjects, in cohort 1 and cohort 2, respectively. Besides, key differential amino acid metabolites, such as choline, creatinine, leucine, tryptophan, and valine, may discriminate MM patients from NC. Moreover, the differential amino acid metabolites were associated with clinical indicators of MM patients.

Conclusions

Our findings indicate that amino acid metabolism disorders are involved in MM. The differential profiles reveal the potential utility of key amino acid metabolites as diagnostic biomarkers of MM. The alterations in metabolome, especially the amino acid metabolome, may provide more evidences for elucidating the pathogenesis and development of MM.

Introduction

Multiple myeloma (MM) is a kind of malignant neoplasm characterized by the accumulation of clonal plasma cells in the bone marrow (Kocemba-Pilarczyk et al., 2018; Xiang et al., 2017). It is the second most common hematologic malignancy (Tabata et al., 2020) and accounts for 1% of all cancers diagnosed in the United States (Schecter & Lentzsch, 2013), with an annual incidence of 6.3 new cases per 100,000 individuals (Neuse et al., 2020). Due to the recurrent relapsing disease course, MM is considered mostly incurable and requires various therapies (Harding et al., 2019). To date, much progress has been made in the treatment strategy of MM. According to these clinical trials (Fayers et al., 2011; Gay et al., 2011; Mateos, 2010), the combination of diverse drugs, such as proteasome inhibitors, immune-modulatory drugs, monoclonal antibodies, HDAC inhibitors and individual CAR-T cell therapy, and autologous hemopoietic stem cell transplantation with traditional drugs, such as corticosteroids, alkylating agents and anthracyclines, has achieved significant clinical effects, but MM remains an incurable disease. Despite considerable advances in treatment, the prognosis of MM is still very heterogeneous (Schavgoulidze et al., 2021). Several reports have demonstrated the links between MM treatment and its early diagnosis (Korde et al., 2015; Mateos et al., 2013). Therefore, a more comprehensive understanding of MM features will aid in the design of an effective treatment strategy for this disease.

Accumulating evidence has confirmed that metabolic alterations are involved in myeloma cell growth and drug resistance (Maiso et al., 2015; Pinto et al., 2020). Metabolomics analysis is a comprehensive method of metabolites that can dynamically monitor the intermediate and final products of biochemical reactions and has been widely used in cancer diagnosis, treatment, and prognosis (Armitage & Southam, 2016; Cao et al., 2020; Kochanowski et al., 2021). By analyzing the metabolic profiles of MM patients at diagnosis and after achieving complete remission, some of the metabolic changes, such as glutamine, cholesterol, and lysine, have been observed, suggesting the potential of metabolic profiles in identifying MM biomarkers or monitoring response to treatment (Puchades-Carrasco et al., 2013). Studies have also revealed significant alterations in amino acid, lipid and energy metabolism by analyzing the metabolomic plasma profile of MM patients and NC and suggested the potential of cellular metabolic processes as promising therapeutic targets in MM (Steiner et al., 2018). In addition, more studies have focused on the indispensable mediator in the plasma microenvironment, amino acids. Glutamine (Cory & Cory, 2006; Pochini et al., 2014) metabolism has proven to have close correlations with hematopoietic malignancies. Du et al. (2018) reported that 23 metabolites changed significantly in MM mainly by arginine, proline and glycerol phospholipid metabolic pathways. Zaal et al. (2017) demonstrated that bortezomib resistance in MM was associated with increased serine synthesis. These findings have confirmed the crucial role of some metabolites and metabolic pathways in the pathogenesis of MM. However, metabolism encompasses the generation of energy, the synthesis and breakdown of glucose, amino acids and fatty acids, and oxidative phosphorylation. The metabolic changes in MM cells include the generation, accumulation and inhibition of metabolites as well as metabolic shifts (El Arfani et al., 2018). Although previous findings have revealed some metabolic changes in MM, the key metabolites and metabolic pathways in MM have not been fully investigated.

Metabolomics approaches can be nontargeted or targeted. Nontargeted metabolomics involves global profiling of the metabolome and often provides more information than targeted metabolomics. However, targeted metabolomics can quantify a select group of metabolites, such as amino acids, which is more quantitative than nontargeted metabolomics (Zhang et al., 2016). In the current study, we conducted nontargeted and targeted metabolomics of amino acids to explore the metabolism disorders and identify metabolomics pathways as well as key differential amino acid metabolites in MM patients. Our findings will provide new insight for elucidating the possible mechanism of MM and exploring promising biomarkers or therapeutic targets for this disease.

Materials and Methods

Study design and patient recruitment

In this study, a total of 70 participants in two cohorts were recruited for metabolomic detection. Patients with MM were collected from Lanzhou University Second Hospital, and NC subjects were recruited through conventional physical examination in the hospital. MM patients were diagnosed according to the criteria of International Myeloma Working Group (IMWG) (Rajkumar, 2016). The International Staging System (ISS) was used to evaluate disease progression (Loehrer, 2006). The patients with MM in the study were all newly diagnosed cases. Subjects with immunodeficiency disease, medication use, smoking habits, or underlying chronic disease were excluded from the study at the preliminary screening stage. All MM patients in this study were not treated with radiotherapy or chemotherapy. Finally, in the cohort 1, 40 participants including 20 MM patients and 20 NC subjects, were selected for nontargeted- and targeted-metabolomics. We also collected the information about clinical variables, including the biochemical detection of serum albumin (ALB), interleukin 6 (IL-6), lactic dehydrogenase (LDH), β2-microglobulin (B2-MG), globulin (GLB), and serum creatinine (SCR) using the Beckman automatic biochemistry analyzer (Beckman Chemistry Analyzer AU5800, Beckman Coulter, Inc., Brea, CA, USA) and Roche electrochemiluminescence system (Cobas e 801 Analyzer, Roche Diagnostics GmbH, Mannheim, Germany), for all the 40 participants in cohort 1. The study of cohort 1 mainly contained two parts: (1) all 40 participants underwent nontargeted metabolomics analysis, and (2) 30 participants (including 15 MM patients and 15 NC subjects) underwent targeted metabolomics analysis. In the cohort 2, another 30 cases, including 15 MM patients and 15 NC subjects, were recruited as the validation group of the targeted metabolomics analysis. Figure 1 summarized the study design, and detailed information on the participants could be found in Table 1 (cohort 1) and Table S1 (cohort 2). This study was approved by the Medical Ethics Committee of Lanzhou University Second Hospital (2018a-037). Written informed consent was obtained from all participants.

Figure 1 Overview of the study design and workflow.

UHPLC-Q-TOF/MS, ultra-high-performance liquid chromatography (UHPLC) with quadrupole time-of-flight mass spectrometry (Q-TOF-MS); MRM-MS, multiple reaction monitoring-mass spectrometry; PCA, principal component analysis; PLS-DA partial-least squares discrimination analysis; OPLS-DA, orthogonal partial least squares discriminant analysis; KEGG, Encyclopedia of Genes and Genomes; ROC, receiver operating characteristic curve.

Nontargeted metabolomics analysis using ultrahigh-performance liquid chromatography (UHPLC) combined with quadrupole time-of-flight mass spectrometry (Q-TOF-MS)

Sample collection and preparation

Peripheral blood samples of all 40 participants in cohort 1, including 20 MM patients and 20 NC subjects, were collected, and serum samples were obtained by centrifugation. The samples were then stored at −80 °C before further processing for UHPLC-Q-TOF/MS analysis.

Table 1 Clinical characteristics of the 40 participants in cohort 1.

Group	Age	Sex	ISS	ALB (g/L)	IL-6 (pg/mL)	LDH (U/L)	β2-MG (ng/mL)	GLB (g/L)	SCR (µmol/L)	
Normal control (NC, n = 20)	55.10 ± 10.43	Female (n = 8)
Male (n = 12)	–	42.16 ± 6.12	4.21 ± 1.98	183.00 ± 66.28	1037.05 ± 423.28	27.62 ± 6.51	67.88 ± 33.47	
Multiple myeloma (MM, n = 20)	54.90 ± 11.29	Female (n = 8)
Male (n = 12)	ISS I (n = 4)
1SS II (n = 6)
ISS III (n = 10)	37.89 ± 9.08	12.24 ± 14.72a∗	241.20 ± 112.80	10,846.45 ± 10,615.85aΔ	49.56 ± 26.42aΔ	243.75 ± 229.60a#	
Notes.

Abbreviations ISS International Staging System

ALB albumin, reference range: 40–55

IL-6 interleukin 6, reference range: 0.00–5.30

LDH lactic dehydrogenase, reference range: 120–250

β2-MG β2-microglobulin, reference range: 609–2366

GLB globulin, reference range: 20–40

SCR serum creatinine, reference range: 41.0–73.0

a Compared with the NC group, ∗Significant association with P < 0.05, #Significant association with P < 0.01, ΔSignificant association with P < 0.001.

UHPLC-Q-TOF-MS analysis

UHPLC analysis was performed with an Agilent 1290 Infinity LC UHPLC system. Chromatographic separation was carried out at 25 °C on a hydrophilic interaction chromatography (HILIC) column with a flow rate of 0.3 mL/min. The mobile phase consisted of solvent A (water + 25 mM ammonium acetate + 25 mM ammonia) and solvent B (acetonitrile). Gradient elution was performed as follows: 95% solvent B with 0–0.5 min; the linear change of solvent B was from 95% to 65% within 0.5–7 min, from 65% to 40% with 7–8 min, and maintained at 40% for 8–9 min; then, the linear change of solvent B was from 40% to 95% within 9–9.1 min and maintained at 95% for 9.1–12 min. The samples were placed in an automatic sampler at 4 °C throughout the whole analysis. To avoid the influence of signal fluctuation caused by instrument detection, continuous analysis of samples was carried out in random order. Moreover, pooled quality control (QC) samples (generated by taking an equal aliquot of all the samples included in the experiment) were run at the beginning of the sample queue for column conditioning and every ten injections thereafter to assess inconsistencies that are particularly evident in large batch acquisitions in terms of retention time drifts and variation in ion intensity over time.

Q-TOF-MS analysis was conducted on an AB Triple TOF 6600 mass spectrometer (AB SCIEX, Framingham, MA, USA). After HILIC chromatographic separation, the electrospray ionization (ESI) source conditions were as follows: ion Source Gas1 (Gas1): 60 psi, Ion Source Gas2 (Gas2): 60, Curtain gas (CUR): 30, source temperature at 600 °C, ionSapary Voltage Floating (ISVF): ±5,500 V (positive and negative model); TOF MS scan m/z range: 60–1,000 Da, product ion scan m/z range: 25–1,000 Da, TOF MS scan accumulation time: 0.20 s/spectra, and product ion scan accumulation time: 0.05 s/spectra. Information-dependent acquisition (IDA) was used to obtain the secondary MS, as well as high sensitivity mode, declustering potential (DP): ±60 V (positive and negative modes), and collision energy: 35 ± 15 eV. IDA was set as exclude isotopes within 4 Da and candidate ions to monitor per cycle: 6.

Data processing and analysis

The raw data were converted into. mzXML format files using the ProteoWizard converter tool and then processed using XCMS software for peak alignment, retention time correction, and peak area extraction. Then, the structures of metabolites were identified by comparing the accuracy of m/z values (<25 ppm) and matching second-stage spectra with the laboratory’s self-built database (Applied Protein Technology Co., Ltd., Shanghai, China).

After data preprocessing by Pareto scaling, multidimensional data analyses were carried out, including unsupervised principal component analysis (PCA), supervised partial least squares discrimination analysis (PLS-DA), and orthogonal partial least squares discriminant analysis (OPLS-DA). Single-dimensional statistical analyses, including Student’s t-test and fold change (FC) analysis, were conducted. The variable importance in the projection (VIP) value (>1) in the PLS-DA model and P value analyzed by Student’s t-test were combined to confirm the significance of differential metabolites. Significant differential metabolites were obtained when VIP > 1 and P value < 0.05, and differential metabolites were selected when VIP > 1 and 0.05 < P value < 0.1. Moreover, Kyoto Encyclopedia of Genes and Genomes (KEGG) pathway (http://www.genome.jp/kegg/) pathway enrichment analysis for significant differential metabolites was conducted by the MetaboAnalyst online tool (http://www.metaboanalyst.ca/MetaboAnalyst/) to investigate the metabolomics pathways.

Targeted metabolomics analysis of amino acids using multiple reaction monitoring-mass spectrometry (MRM-MS)

Sample collection and preparation

Peripheral blood samples of 30 out of 40 participants in cohort 1 (including 15 MM patients and 15 NC subjects) and another 30 cases in cohort 2 (the validation group, including 15 MM patients and 15 NC subjects) were collected for targeted metabolomics analysis. Serum samples were obtained by centrifuging at 1, 000 × g for 10 min and then stored at −80 °C.

MRM-MS analysis

The analysis was performed with an Agilent 1290 Infinity LC UHPLC system. The mobile phases were as follows: solvent A was 25 mM ammonium formate + 0.08% formic acid (FA) in water, and solvent B was 0.1% FA-acetonitrile. The sample was placed in an automatic sampler at 4 °C, the column temperature was 40 °C, the flow rate was 250 µL/min, and the injection volume was 1 µL. Gradient elution was performed as follows: the linear change of solvent B was from 90% to 70% within 0–12 min; from 70% to 50% with 12–18 min; from 50% to 40% within 18–25 min, from 40% to 90% with 30–30.1 min; and maintained at 90% for 30.1–37 min. In the sample queue, a QC sample was also set in a certain number of experimental samples at every interval to test and evaluate the stability and repeatability of the system. MS analysis was carried out in positive ionization mode on a 5500 QTRAP mass spectrometer (AB SCIEX, Framingham, MA, USA). The conditions of the 5500 QTRAP ESI source were as follows: ion source temperature, 500 °C; Gas1, 40 psi; Gas2, 40 psi; CUR, 30 psi; and ISVF, 5500 V. MRM mode was then used to detect the ion pairs.

Data processing and analysis

The chromatographic peak area and analyte retention time were extracted using Multiquant software. Based on the standards of amino acids and their derivatives, the analyte retention time was corrected, followed by identification of the metabolites. PLS-DA was also conducted to explore the amino acids profiling and differential amino acid metabolites between groups. Score scatter plots and loading plots were generated to visualize the separation of samples and metabolites. MedCalc software (v19.0.4) was utilized for receiver operating characteristic curve (ROC) analysis to evaluate the diagnostic accuracy of amino acid metabolites. Moreover, Spearman correlation analysis was performed with the statistical platform R package (v3.2.4). KEGG pathway enrichment analysis for significant differential metabolites was also conducted by the MetaboAnalyst online tool.

Results

Nontargeted metabolomics analysis using UHPLC-Q-TOF-MS

Multivariate statistical analysis

To analyze the metabolic changes between MM patients and NC, nontargeted metabolomics analysis was carried out using UHPLC-Q-TOF-MS. As a result, the PCA score plots showed a clear separation between MM patients and NC in both positive-ion and negative-ion modes (Fig. 2A). To further identify ion peaks that could possibly be used to differentiate the metabolic profiles of MM patients and NC, the supervised PLS-DA model was conducted. The results showed that MM patients were also separated from NC by the PLS-DA score plots in both modes (Fig. 2B). Consistent results were also obtained by the OPLS-DA score plots in both modes (Fig. 2C). Further permutation tests consisting of 200 permutations demonstrated that the model was not overfitted (positive-ion model: R2 = (0.0, 0.707), Q2 = (0.0, −0.3673); negative-ion model: R2 = (0.0, 0.8331), Q2 = (0.0, −0.4174); Fig. 2D). These data indicated that there was a significant metabolic change between MM patients and NC.

Figure 2 Score plots of supervised PCA, PLS-DA, and OPLS-DA analysis based on the combination of UHPLC-Q-TOF/MS data from MM patients and NC.

(A) PCA score plots between MM patients and normal controls (NC) in both positive-ion and negative-ion modes. (B) PLS-DA score plots between MM patients and normal controls (NC) in both modes. (C) OPLS-DA score plots between MM patients and normal controls (NC) in both modes. (D) Permutation tests consisting of 200 permutations demonstrated that the OPLS-DA model was not overfitted in either mode.

Analysis of differential metabolites between MM patients and NC

To compare differences in metabolites between MM patients and NC, differential metabolites were screened out between MM patients and NC in both positive-ion and negative-ion modes based on VIP >1 in the PLS-DA model and P < 0.1 analyzed by Student’s t-test. The results showed that a total of 62 differential metabolites were identified between MM patients and NC in positive-ion mode, of which 56 were significant (P < 0.05). Similarly, 45 differential metabolites were identified between MM patients and NC in negative-ion mode, of which 38 were significant (P < 0.05) (Table S2). These data confirmed that there were significant differences in metabolites between MM patients and NC.

Metabolic pathway analysis

To investigate the metabolomic pathways involved in MM development, the differential abundant metabolites were enriched for the related metabolic pathway analysis using the MetaboAnalyst online tool. As shown in Fig. 3, differential metabolites were significantly enriched in multiple metabolic pathways, such as protein digestion and absorption (including L-alanine, L-glutamine, L-tryptophan, L-histidine, L-asparagine, L-valine, and L-isoleucine), ABC transporters (including L-alanine, L-glutamine, L-histidine, L-valine, and L-isoleucine), and biosynthesis of amino acids (L-alanine, L-glutamine, L-tryptophan, L-histidine, L-asparagine, L-valine, L-citrulline, and L-isoleucine). Notably, these pathways were associated with amino acid metabolism.

Figure 3 Metabolic pathway enrichment analysis of differential metabolites between MM patients and NC.

Each related metabolic pathway is shown as a circle, whose size and color are based on the pathway impact value and the P value, respectively.

Targeted metabolomics analysis of amino acids using MRM-MS

Serum amino acid metabolites of MM patients and NC showed distinguished profiles

Previously, the results indicated that the altered metabolic profile of MM was mainly involved in amino acid metabolomics features. We further conducted a target amino acid metabolomics analysis to gain more insight into MM metabolism. Generally, in the cases of cohort 1, the PLS-DA two-dimensional (2D) score plot showed clearly distinguished profiles between the MM and NC groups (Fig. 4A). Moreover, scatter loading plot analysis was carried out to evaluate whether these differential amino acid metabolites accounted for the distinguished profiles. MM patients exhibited higher levels of choline, creatinine, glutamate, and asparagine as well as lower levels of alanine/sarcosine, valine, tryptophan, and cystine than NC (Fig. 4B). These data suggested that altered amino acid profiles could discriminate MM patients from NC.

Figure 4 PLS-DA analysis based on MRM-MS data and functional pathway of key differential metabolites between MM and NC groups.

(A, B) PLS-DA scores and loading plots between MM patients and normal controls based on the MRM-MS data. (C, D) Metabolic pathway enrichment analysis of differential amino acids metabolites between MM patients and NC. In (C), each related metabolic pathway is shown as a circle, whose size and color are based on the pathway impact value and the P value, respectively.

Expression of differential amino acid metabolites and the KEGG functional analysis

According to the metabolomics features, with the purpose of finding potential diagnostic biomarkers or therapeutic targets for MM, we focused on the differential amino acid metabolites in MM patients compared to those in NC. To further investigate the detailed variety degree of each metabolite, we analyzed the intensity differences in a total of 28 amino acid metabolites. Compared to NC, MM patients exhibited differential metabolic profiles. Overall, in the cases of cohort 1, there was a total of 12 significantly differential abundant amino acid metabolites in MM patients relative to NC (P < 0.05), including lysine, leucine, isoleucine, histidine, valine, threonine, glutamine, tryptophan, choline, ornithine, creatinine, and alanine/sarcosine (Table S3). Furthermore, we validated these findings with the samples in cohort 2, and the results showed similar dysregulation trends with the findings of cohort 1, that we identified 14 significantly differential abundant amino acid metabolites in MM group (Table S4). Specifically, the high expression levels of creatinine (FC = 2.029, P < 0.01 in cohort 1; FC = 1.632, P < 0.05 in cohort 2) and choline (FC = 1.508, P < 0.001 in cohort 1; FC = 1.234, P = 0.125 in cohort 2) were found in MM patients in both cohort 1 and cohort 2. Besides, since glutamate signaling has been reported involved in malignant disorders (Willard & Koochekpour, 2013), we also examined glutamate levels and found slightly high levels in MM patients (FC = 1.186, P = 0.245 in cohort 1; FC = 1.292, P = 0.346 in cohort 2).

To determine the pathway impact of these differential amino acid metabolites, we performed pathway classifications provided by KEGG analysis. Results indicated that the 12 differential abundant metabolites in cohort 1 were significantly enriched in 22 metabolic pathways, such as aminoacyl-tRNA biosynthesis, valine/leucine/isoleucine biosynthesis, phenylalanine/tyrosine/tryptophan biosynthesis, D-glutamine/D-glutamate metabolism, and histidine metabolism (Figs. 4C, 4D).

Differential amino acid metabolites may discriminate MM patients from NC

We subsequently evaluated the performances of these differential abundant metabolites in distinguishing MM from NC group. ROC analysis showed that, the amino acid metabolites classifier could act as potential disease diagnostic biomarkers in both cohort 1 and cohort 2 (AUC > 0.7, P < 0.05, Table 2 and Fig. S1). In MM patients of cohort 1, the AUC values of two representative upregulated amino acid metabolites, choline and creatinine, were 0.822 and 0.822, respectively (Table 2). Similarly, the AUC of creatinine in the cohort 2 was 0.747 (Table 2). Meanwhile, the ROC analysis also showed good diagnostic values of the downregulated amino acid metabolites in MM, such as leucine (AUC = 0.871 in cohort 1, AUC = 0.920 in cohort 2) and tryptophan (AUC = 0.884 in cohort 1, AUC = 0.902 in cohort 2). Specifically, we observed extremely high AUC values of the metabolite valine in both cohort 1 (AUC = 0.964) and cohort 2 (AUC = 0.960). The results above suggested that these key differential amino acid metabolites might be regarded as potential biomarkers for the diagnosis or therapeutic targets of MM.

Table 2 Performance of differential metabolites in distinguishing MM from the NC group.

Cohort 1	Cohort 2	
Metabolites	Sensitivity	Specificity	AUC	P	Metabolites	Sensitivity	Specificity	AUC	P	
Alanine/sarcosine	60.0	93.3	0.751	0.008	alanine/sarcosine	66.7	93.3	0.773	0.003	
Choline	66.7	100.0	0.822	<0.001	creatinine	73.3	86.7	0.747	0.012	
Creatinine	66.7	100.0	0.822	<0.001	Glutamine	66.7	86.7	0.773	0.003	
Glutamine	100.0	60.0	0.778	0.002	histidine	80.0	86.7	0.907	<0.001	
histidine	80.0	80.0	0.778	0.002	Isoleucine	60.0	93.3	0.840	<0.001	
Isoleucine	73.3	80.0	0.782	0.001	leucine	86.7	86.7	0.920	<0.001	
Leucine	93.3	73.3	0.871	<0.001	lysine	40.0	100.0	0.724	0.017	
Lysine	40.0	100.0	0.720	0.024	ornithine	46.7	100.0	0.769	0.002	
Ornithine	66.7	80.0	0.782	0.001	serine	80.0	73.3	0.738	0.012	
Threonine	93.3	73.3	0.853	<0.001	taurine	86.7	60.0	0.764	0.003	
Tryptophan	73.3	93.3	0.884	<0.001	threonine	93.3	66.7	0.827	<0.001	
Valine	86.7	93.3	0.964	<0.001	tryptophan	73.3	93.3	0.902	<0.001	
					tyrosine	80.0	66.7	0.769	0.002	
					valine	86.7	93.3	0.960	<0.001	

Differential amino acid metabolites were associated with MM phenotypes

To investigate the correlations between differential amino acid metabolites and MM-associated phenotypes, we analyzed the correlations among 6 clinical indices/variables and 12 amino acid metabolites that differed significantly in abundance between MM patients and NC in the cohort 1 (Fig. 5). The clinical variables included serum albumin (ALB), interleukin 6 (IL-6), lactic dehydrogenase (LDH), β2-microglobulin (B2-MG), globulin (GLB), and serum creatinine (SCR). Among 12 amino acid metabolites whose abundances differed in MM patients, two mainly correlation clusters were obtained. The cluster of upregulated components, creatinine and choline in MM patients, was positively correlated with the levels of LDH, SCR, and B2-MG. Moreover, in the downregulated amino acids cluster, 10 amino acid metabolites were negatively correlated with the levels of IL-6 and GLB. Among these metabolites, the relationship between GLB and valine/isoleucine/tryptophan/leucine/sarcosine exhibited high correlation coefficients (adjusted P < 0.01). These data suggested that the combination of clinical indices and amino acid metabolites alterations facilitate a better understanding of MM progression.

Figure 5 Correlation analysis of 12 differential amino acids identified from MM patients vs. NC with six clinical indicators.

Numbers in the lower left panel: value of the correlation coefficient; symbols in the upper right panel: results of the significance test; *adjusted P < 0.05, ** adjusted P < 0.01. ALB, albumin; LDH, lactic dehydrogenase; B2-MG, β2-microglobulin.

Discussion

Metabolite disorder plays an essential role in cancer initiation and progression (La Vecchia & Sebastián, 2020; Whisner & Aktipis, 2019). Metabolic changes in the tumor microenvironment affect the effects of immunotherapy (Kouidhi, Ben Ayed & Benammar Elgaaied, 2018). Therefore, a better understanding of metabolic changes in the tumor microenvironment will improve the beneficial effects of immunotherapy. Recently, with the development of metabolomics, more researchers have focused on the clinical examination of metabolite variation, which might provide more predictors and clues in disease diagnosis and treatment (Chen et al., 2019).

MM, as a heterogenic disease with dynamic metabolic processes in bone marrow and its microenvironment, has been reported to exhibit metabolic changes (El Arfani et al., 2018; Silva et al., 2020). Despite this, the specific metabolic profiles in MM patients and potential clinical biomarkers remain ambiguous and need to be further clarified. Metabolomics analysis is a powerful means for investigating metabolic processes, revealing metabolic mechanisms and identifying crucial biomarkers responsible for metabolic alteration. In the present study, we conducted nontargeted metabolomics analysis using UHPLC-Q-TOF-MS and found a significant metabolic change between MM patients and NC. Differential metabolites could discriminate MM patients and NC, which were significantly enriched in metabolic pathways related to amino acid metabolism, such as protein digestion, and absorption and biosynthesis of amino acids. These data suggested the key role of amino acid metabolism in MM. We then conducted the targeted metabolomics analysis of amino acids using MRM-MS in two cohorts of participants and found that serum amino acid metabolites of MM patients and NC showed distinguished profiles. Compared to NC, MM patients exhibited higher levels of choline and creatinine as well as lower levels of valine, tryptophan, leucine, etc. In another related study, the lipids profiles of MM showed lower concentrations of phosphatidylcholine (PC), lysophosphatidylcholine (LPC) and sphingomyelins (SM) (Silva et al., 2020). Besides, the concentration of essential amino acids, especially tryptophan, was significantly decreased in MM cases (Silva et al., 2020), which was consistent with our findings. Further ROC analysis in this study showed relatively good diagnostic values of upregulated amino acid metabolites, including choline and creatinine, and downregulated amino acid metabolites, such as leucine, tryptophan and valine. These results suggested that the key differential amino acid metabolites could be used as promising biomarkers for MM diagnosis. In addition, the clinical indicators were found to be related to altered amino acid metabolomics in correlation analysis. MM patients were accompanied by elevation of β2-microglobulin (β2-MG), and researchers found an inverse correlation between the concentration of tryptophan and β2-MG (Silva et al., 2020). In the current study, we also observed a negative correlation between the differential metabolite tryptophan and β2-MG. Besides, the downregulated amino acids, including lysine, leucine, and valine, were also negatively related with the high level of β2- MG. Taken together, we hypothesized that the diagnostic value may be improved by combining examination of clinical indicators and amino acid levels.

Amino acids and proteins have been shown to play central roles in cellular metabolism (Rizzieri, Paul & Kang, 2019; Zhang et al., 2019). Amino acids are responsible for the formation of a variety of components that are used for cell proliferation (Martinez-Outschoorn et al., 2017). Notably, it has been reported that targeting amino acid metabolism may be a promising therapy for cancer, suggesting the crucial role of amino acid metabolism in cancer (Bai et al., 2020; Boon et al., 2020; Wang & Zou, 2020). Moreover, protein digestion and absorption is also an enriched metabolism/digestive pathway involved in mechanisms of inhibitory effect of 3-O-kaempferol-3-O-acetyl-6-O-(p-coumaroyl)- β-D-glucopyranoside (HK-11, a flavonoid compound) on MM cell proliferation (Hou et al., 2020), implying the potential role of protein digestion and absorption pathway in MM development. Our nontarget metabolomics analysis revealed that the differential metabolites between MM patients and NC were significantly enriched in metabolic pathways related to amino acid metabolism. It can therefore be speculated that amino acid metabolism is involved in MM. Further targeted metabolomics analysis of amino acids showed that the concentration of key metabolites was altered in MM, and most of the amino acid metabolites was downregulated. We highlight that these key metabolites may act as potential biomarkers for MM, and that the supplementation of the disease phenotype-negatively related metabolic components may help prevent or improve the prognosis of MM. Recent studies have also investigated the proteomic alterations in the bone marrow interstitial fluid and serum samples, and identified candidate proteins to be associated with drug resistance in MM patients (Chanukuppa et al., 2019; Chanukuppa et al., 2021). The complex metabolite and lipid profiling of bone marrow plasma could help differentiate patients with monoclonal gammopathy of undetermined significance (MGUS) from MM (Gonsalves et al., 2020). Therefore, the metabolic and proteomic detection provides more evidences for the development of improved diagnosis and treatment of MM. In the current study, we also conducted the targeted metabolomics analysis involving amino acids in MM patients, and it’s necessary to broaden the metabolic classes to find novel biomarkers in the future.

In conclusion, our results reveal that amino acid metabolism disorder is involved in MM. Key differential amino acid metabolites, such as choline, creatinine, leucine, tryptophan, and valine, may discriminate MM patients from NC. The alteration in metabolism, especially amino acid metabolism, may provide evidences for elucidating the pathogenesis of MM and lay a reference for clinical diagnosis and therapy of this disease. However, there are still several limitations in our research. Although we summarized differential amino acid metabolites and explored their diagnostic value, more functional validations in vivo are still needed. Moreover, the sample size was relatively small to make some general conclusions. More research is needed to support the results and to investigate the underlying biological functions of the key amino acid metabolites with large scale and mechanism study in the future. Besides, confounders such as medication taken or smoking habits of the individuals can influence the metabolic trend, comprehensive research involving the factors should be conducted in larger cohort in future research.

Supplemental Information

Supplemental Information 1 ROC curve analysis of the diagnostic value of differential amino acids for MM in cohort 1 (A) and cohort 2 (B)

Click here for additional data file.

Supplemental Information 2 Clinical characteristics of the 30 participants in cohort 2

Abbreviations: ISS, International Staging System.

Click here for additional data file.

Supplemental Information 3 Differential metabolites between MM patients and healthy controls

Click here for additional data file.

Supplemental Information 4 Differential amino acids metabolites between MM patients and healthy controls (cohort 1)

Click here for additional data file.

Supplemental Information 5 Differential amino acids metabolites between MM patients and healthy controls in the validation group (cohort 2)

Click here for additional data file.

Additional Information and Declarations

Competing Interests

Author Contributions

Human Ethics

Data Availability

The authors declare there are no competing interests.

Lingling Yue conceived and designed the experiments, performed the experiments, prepared figures and/or tables, authored or reviewed drafts of the paper, and approved the final draft.

Pengyun Zeng performed the experiments, analyzed the data, prepared figures and/or tables, and approved the final draft.

Yanhong Li performed the experiments, analyzed the data, authored or reviewed drafts of the paper, and approved the final draft.

Ye Chai analyzed the data, authored or reviewed drafts of the paper, and approved the final draft.

Chongyang Wu analyzed the data, prepared figures and/or tables, and approved the final draft.

Bingren Gao conceived and designed the experiments, analyzed the data, prepared figures and/or tables, authored or reviewed drafts of the paper, and approved the final draft.

The following information was supplied relating to ethical approvals (i.e., approving body and any reference numbers):

This study was approved by the Medical Ethics Committee of Lanzhou University Second Hospital (2018a-037).

The following information was supplied regarding data availability:

The raw data of this study are available in FigShare: Gao, Bingren; Yue, Lingling (2021): Early Warning Research of Multiple Myeloma Based on LC-MSMS Technology and Amino Metabolism. figshare. Dataset. https://doi.org/10.6084/m9.figshare.14904135.v1.

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
