# Peer review of "Nontargeted and targeted metabolomics approaches reveal the key amino acid alterations involved in multiple myeloma"

_PeerJ, doi:10.7717/peerj.12918_

## Round 0.1 · original submission · Major Revisions

As the reviews reveal, the referees discovered a number of conceptual, methodological and legibility flaws. Major concerns are inadequate discussion of findings and inappropriate presentation of data. The manuscript is therefore recommended to revise the manuscript per reviewers' comments and submit a point-to-point response to reviewers' comments for further editorial consideration.

Reviewer 1 ·

Basic reporting

The manuscript is written in proper professional English, clear and unambiguous. The introduction describes well enough the background of the study and included sufficient references. However, there is no message of the novelty of the study compared to the similar studies previously published.

Experimental design

- Number of sample is to less for a conclusion. Validation study with higher sample number is recommended.
- The individual stratification is not clearly described in the manuscript.
- There is no information about cofounders, e.g., medication, comorbidity, smoking, etc.
- Detection and annotation of broaden metabolic groups from the nontargeted metabolomics will be beneficial.
- QC samples used for nontargeted metabolomics should be clearly defined.
- Information on how the system suitability is assessed is missing.

Validity of the findings

Validation study is recommended as number of sample used in this study is too low.

Reviewer 2 ·

Basic reporting

This manuscript by Yue et al. uses nontargeted and targeted metabolomics approaches to investigate possible metabolic differences between 20 newly diagnoses MM patients and 20 normal controls. They explored the differences in amino acid metabolites between the two groups and claim that their findings will provide” new insight for elucidating the possible mechanism of MM and exploring promising biomarkers or therapeutic targets for this disease”.
Based on previous studies in this area, these results are not new because many authors have already showed amino acid alterations in MM patients (references already cited in introduction) when compared to normal controls. An important issue currently is if these alterations are related to the tumor or are in born errors that predispose to MM development. The reference Biochemical phenotyping of multiple myeloma patients at diagnosis reveals a disorder of mitochondrial complexes I and II and a Hartnup-like disturbance as underlying conditions, also influencing different stages of the disease. da Silva IDCG, de Castro Levatti EV, Pedroso AP, Marchioni DML, Carioca AAF, Colleoni GWB.Sci Rep. 2020 Dec 14;10(1):21836. doi: 10.1038/s41598-020-75862-4. was missed and can be useful to improve this discussion.

Worldwide, MM diagnosis is based in laboratorial criteria. Therefore, it is hard to believe that metabolomic profile could be used as a more efficient biomarker than the currently used.
The authors should also develop the argument of how their findings could be therapeutic targets for this incurable disease.
Minor points:
Summary: Please review the entire text to include the term "amino acid metabolites" rather than just amino acids when referring to choline, creatine and creatinine.
Incorrect: “ROC curve analysis showed the high diagnostic accuracy of key amino acids, including choline, creatinine, tryptophan, and valine. “
Correct: “Key differential amino acid metabolites, such as choline, creatinine, tryptophan, and valine, may discriminate MM patients from NC. “
Correct: “The changes in amino acid metabolites may provide some clues for elucidating the pathogenesis of MM.”
Introduction: “and there are approximately 21,000 new diagnoses each year”. Please specify where. China? US? World?
“Although there have been some achievements in MM remission, several reports have demonstrated that the real success of MM treatment depends on its early diagnosis.” Please review this concept. Genetic-molecular background associated to clinical staging (R-ISI) are currently considered the gold standard to define prognosis of MM patients.
“The metabolic changes in MMC cells include the generation, accumulation and inhibition of metabolites as well as metabolic shifts”. Please correct MMC to MM.
Figure 1: Please change for “20 normal controls”.
Raw data was shared.
Professional English was used.

Experimental design

This manuscript by Yue et al. uses nontargeted and targeted metabolomics approaches to investigate possible metabolic differences between 20 newly diagnoses MM patients and 20 normal controls. They explored the differences in amino acid metabolites between the two groups and claim that their findings will provide” new insight for elucidating the possible mechanism of MM and exploring promising biomarkers or therapeutic targets for this disease”.
Based on previous studies in this area, these results are not new because many authors have already showed amino acid alterations in MM patients (references already cited in introduction) when compared to normal controls.

Validity of the findings

This manuscript by Yue et al. uses nontargeted and targeted metabolomics approaches to investigate possible metabolic differences between 20 newly diagnoses MM patients and 20 normal controls. They explored the differences in amino acid metabolites between the two groups and claim that their findings will provide” new insight for elucidating the possible mechanism of MM and exploring promising biomarkers or therapeutic targets for this disease”.
Based on previous studies in this area, these results are not new because many authors have already showed amino acid alterations in MM patients (references already cited in introduction) when compared to normal controls.
Worldwide, MM diagnosis is based in laboratorial criteria. Therefore, it is hard to believe that metabolomic profile could be used as a more efficient biomarker than the currently used.
The authors should also develop the argument of how their findings could be therapeutic targets for this incurable disease.

---

## Round 0.2 · Major Revisions

As you can see from the referee report, the expert reviewer has agreed on the potential of the work but reviewer also reported the substantial methodological, scope and legibility issues, and I concur. A major criticism is to broaden the scope with additional analysis, requisite evidence to justify the claims and inadequate discussion of results. Additionally, careful proofreading and professional language editing is encouraged for further improvement. A resubmission is commended with point-to-point response to reviewer comments for this manuscript to be meritorious and considerable potential interest.

Reviewer 1 ·

Basic reporting

In the manuscript the authors reported their observation on regulation of some amino acids in multiple myeloma patients in comparation to that in normal controls. The manuscript is clearly written with proper professional English. The literature references are appropriately referenced in this manuscript.

Experimental design

In the manuscript there is no data about confounders, e.g., medications taken by the individuals, smoking habits, etc. Confounders can influence the metabolic trend. It must be reported and incorporated in the data analysis.

In materials and methods part the authors didn't mention about type of QC samples applied. The labelled IS should be described more detail about the mass ranges and retention times.

Validity of the findings

There are already some published data on metabolomics of multiple myeloma, including data on amino acids, e.g., Chanukuppa et al, 2019; Gonsalves et al, 2020. I suggest not to restrict the observation on amino acids instead broaden the metabolic classes to find novel biomarkers.

Additional comments

In the manuscript the authors reported their observation on regulation of some amino acids in multiple myeloma patients in comparation to that in normal controls. However, there are already some published data on metabolomics of multiple myeloma, including data on amino acids, e.g., Chanukuppa et al, 2019; Gonsalves et al, 2020. I suggest not to restrict the observation on amino acids instead broaden the metabolic classes to find novel biomarkers.

In the manuscript there is no data about confounders, e.g., medications taken by the individuals, smoking habits, etc. Confounders can influence the metabolic trend. It must be reported and incorporated in the data analysis.

In materials and methods part the authors didn't mention about type of QC samples applied. Please elaborate this in the manuscript. The labelled IS should be described more detail about the mass ranges and retention times. For the metabolic annotation for nontargeted metabolomics, did you use retention time or retention index?

---

## Round 0.3 · Minor Revisions

The reviewer acknowledged that the article had improved greatly, but also advised small modifications. That is something I agree with. Please provide a point-by-point response to reviewer comments for future editorial consideration.

Reviewer 1 ·

Basic reporting

The manuscript is written in proper professional English. The introduction is well written with sufficient field background and references.

Experimental design

Research question is meaningful. Methods is decently explained, however, I miss the information about biochemical annotation on nontargeted metabolomics part. Also, the authors didn't mention confidence level of the annotation.
Please provide these information.

Validity of the findings

The authors have responded to my concern in the first round of reviewing process.

Additional comments

N/A

---

## Round 0.4 · accepted · Accept

The current revision and the response to reviewer comments are acceptable.